


# Topographic Effects of Svalbard on Warm and Moist Air Intrusions into the Central Arctic

Jan Landwehrs[1], Sonja Murto[2,3], Florian Gebhardt[1], Ella Gilbert[4], and Annette Rinke[1]

[1]Alfred Wegener Institute, Helmholtz Centre for Polar and Marine Research, Telegrafenberg A45, Potsdam, Germany
[2]Stockholm University and Bolin Centre for Climate Research, Department of Meteorology, Stockholm, Sweden
[3]Department of Earth Sciences, Uppsala University, Uppsala, Sweden
[4]British Antarctic Survey, Madingley Road, Cambridge, UK

**Correspondence:** Jan Landwehrs (jan.landwehrs@awi.de)

**Abstract.** Warm air intrusions (WAIs) along the North Atlantic pathway are key drivers of warm extremes in the central Arctic. The Svalbard archipelago acts as a major topographic barrier in the middle of this gateway, but its role in modulating WAIs and their impacts has not been studied in detail. We combine (i) high-resolution regional ICON simulations with and without Svalbard's topography, (ii) Lagrangian back-trajectories, and (iii) observations from the MOSAiC expedition to analyze a strong WAI event in mid-April 2020, and extend the analysis with (iv) climatological composites from an ICON simulation for 2000–2022. Based on the April 2020 case study, we show that Svalbard's influence can extend ~500 km downstream over sea ice and was observed near 84° N during MOSAiC. The response depends on the static stability of the impinging flow: stable conditions favor flow-around with barrier and gap winds and a broad lee wake, leading to downstream reductions in wind speed (by $>5\,\mathrm{m\,s^{-1}}$), near-surface temperature ($>3\,\mathrm{K}$), and column-integrated water vapor ($>1\,\mathrm{kg\,m^{-2}}$). Under less stable flow-over conditions, föhn signatures yield lower-tropospheric warming ($>1\,\mathrm{K}$) and drying, reduced low-level cloud cover ($>20\,\%$), and decreased (increased) downwelling longwave (shortwave) radiation ($>20\,\mathrm{W\,m^{-2}}$). Springtime composites reveal that these signals recur during southerly advection events, can extend several hundred kilometers into the central Arctic, and vary in character with poleward wind speed, moisture transport, and static stability linked to the synoptic situation. Together, the results demonstrate that Svalbard's topography systematically modulates the dynamical and thermodynamic imprint of WAIs, with effects detectable far downstream in both model experiments and MOSAiC observations.

## 1 Introduction

The Arctic is warming much faster than the globe ("Arctic Amplification", Rantanen et al., 2022) and poleward atmospheric transport of heat and moisture is a key contributor to that (Serreze and Barry, 2011; Previdi et al., 2021; Taylor et al., 2022; Wendisch et al., 2023). In this context, moist and warm air intrusions (WAIs) are impactful synoptic events. They are closely linked to warm extremes (e.g., Messori et al., 2018; Cardinale and Rose, 2022; Scholz and Lora, 2024) and positive anomalies of the surface energy budget (Papritz et al., 2023; Murto et al., 2023), and are characterized by strong meridional advection of warm, often moist, air masses from lower latitudes that penetrate deep into the Arctic (e.g., Woods and Caballero, 2016; Pithan et al., 2018; You et al., 2022; Kolbe et al., 2023). Many studies use the closely related concept of atmospheric rivers (ARs) to



study similar events and their impacts (Ma et al., 2021, 2024a, b; Zhang et al., 2023a, b; Wang et al., 2024; Thaker et al., 2025).
Here, we refer to these events collectively as WAIs, and consider ARs a subset that meets the specific criteria commonly used
in AR detection (e.g., a high length-to-width ratio).

One of the primary WAI pathway into the central Arctic is the northern North Atlantic (Woods and Caballero, 2016; Nygård
et al., 2020; Papritz and Dunn-Sigouin, 2020; Papritz et al., 2022, 2023), with Svalbard situated in the center of this gateway.
Observational and modeling studies show that WAIs can strongly impact the surface energy fluxes and resulting warming when
passing Ny-Ålesund in western Svalbard (Dahlke and Maturilli, 2017; Yamanouchi, 2019; Bresson et al., 2022). This study
instead shifts the perspective to examine how Svalbard itself modulates WAIs and their downstream evolution, acting as a major
topographic barrier with mountain ranges reaching ∼1700 m height. Whether air masses pass over or divert around Svalbard
depends on the linearity of the flow regime, often summarized by the non-dimensional mountain height $\hat{h} = Nh/U$ (inverse
of the Froude number $Fr$; $N$ Brunt-Väisälä frequency, $h$ characteristic mountain height, $U$ wind component perpendicular to
the barrier Elvidge et al., 2016). For $\hat{h} < 1$ (weak stability / strong wind) linear flow-over dominates, while in a non-linear
regime with $\hat{h} > 1$ (strong stability / low wind) upstream blocking yields flow-around with barrier and gap winds and a broad
lee wake (Elvidge et al., 2016). In non-linear regimes, high-amplitude gravity waves and their breaking over the mountains can
cause turbulent dissipation of kinetic energy and deceleration of the cross-barrier flow. This leads to strong lee-side downslope
windstorms and a hydraulic jump where the flow transitions from a supercritical back to a subcritical regime, accompanied
by an abrupt deceleration and deepening of the flow (Elvidge et al., 2016; Shestakova et al., 2022). Föhn is typically marked
by lee-side warming, drying and cloud clearance, yielding enhanced (reduced) downwelling shortwave (longwave) radiation.
Following Elvidge and Renfrew (2016), four processes can contribute to this, depending on the flow regime: (i) isentropic
drawdown of warm and dry air from aloft, (ii) latent heat release and precipitation on the windward slope, (iii) mechanical
mixing that entrains warmer air from above into the föhn flow, and (iv) radiative warming associated with cloud clearance. In
a non-linear flow regime, föhn warming is expected mainly in the immediate lee before the hydraulic jump deflects the warm
föhn flow upwards. In a linear regime, the föhn warming can propagate downstream over longer distances (Elvidge et al., 2016;
Shestakova et al., 2022).

Föhn winds, particularly if associated with WAIs, have been linked to extreme surface melt on glaciers and ice sheets in the
Arctic, including Novaya Zemlya (Haacker et al., 2024) and northeast Greenland (Mattingly et al., 2023). Such melt extremes
are important drivers for glacier ice mass loss and consequently sea level rise. Related topographic effects in the polar regions,
including the role of föhn in modulating surface energy balance and melt, are well documented for Antarctica and especially
the Antarctic Peninsula (e.g., Elvidge et al., 2016; Elvidge and Renfrew, 2016; Gilbert et al., 2022a, b; Orr et al., 2021, 2023;
Lu et al., 2023).

For Svalbard, a few studies examined topographic and föhn effects during episodes of easterly flow (e.g., Dörnbrack et al.,
2010; Kilpeläinen et al., 2011; Reeve and Kolstad, 2011; Shestakova et al., 2022). Of those, Shestakova et al. (2022) present
the most recent and comprehensive case study of topographic effects in the Arctic. During 30–31 May 2017, they observe both
a quasi-linear flow regime with high incoming wind speed and close to neutral stratification ($Fr \approx 1$) followed by a period
of more non-linear flow ($Fr < 0.8$) over Svalbard. The latter case showed amplified downslope windstorms, gap winds and





wake formation downstream of the hydraulic jump. Shestakova et al. (2022) showed that föhn warming (up to 10 K) and drying

occurred over Svalbard's western slopes but also propagated more than 100 km downstream. The study identified isentropic drawdown as the main föhn mechanism. It was found that the föhn accelerated snow melt primarily via increased downward shortwave radiation and sensible heat flux at the surface. There was no clear correlation between the non-linearity of the flow and the horizontal range of the föhn warming, possibly owing to relatively small variations in the Froude number as well as the complex orography (Shestakova et al., 2022).

The role of Svalbard's topography in southerly warm and moist advection events remains poorly understood, despite the fact that WAIs along the North Atlantic pathway are critical drivers of warm extremes and climate change in the central Arctic. Here we address this gap by characterizing the main dynamical and thermodynamical effects of Svalbard's topography during southerly advection, as well as their downstream extent and their sensitivity to parameters of the flow regime. To this end, we combine high-resolution regional ICON model simulations with unique observations from the central Arctic collected during

the MOSAiC expedition aboard the RV Polarstern (Shupe et al., 2020). We analyze an exceptional WAI event that followed the North Atlantic pathway, crossed Svalbard, and reached MOSAiC around 14–16 April 2020 about 500 km farther north (Rinke et al., 2021; Pithan et al., 2023; Kirbus et al., 2023; Svensson et al., 2023). This provides a rare opportunity to test whether topographic effects of Svalbard on WAIs extend that far north. To isolate these effects, we conduct paired ICON model experiments with and without Svalbard's orography and analyze Lagrangian backward trajectories initialized around

RV Polarstern. Beyond this case study, we place our findings in a climatological context by analyzing southerly advection events across Svalbard during springtime in a pan-Arctic ICON climate simulation for 2000–2022. Compared with available reanalyses like ERA5 (Hersbach et al., 2020) or CARRA (Køltzow et al., 2022), the ICON simulations together offer an optimal combination of (spatial and temporal) resolution and coverage to resolve topographic effects at both case-study and climatological scales.

This study presents the first detailed investigation of how Svalbard's topography modulates WAIs into the central Arctic. Section 2 outlines the ICON model simulations and LAGRANTO trajectory analyses. Section 3.1 presents results for the mid-April 2020 case, and Sect. 3.2 discusses climatological composites for spring 2000–2022. Finally, Section 4 summarizes the main findings.

## 2   Methods & Data

### 2.1   ICON simulations for the April 2020 WAI case study

Our main tool is the ICON atmospheric model (Zängl et al., 2015). Simulations for the April 2020 WAI case study use the open-source release ICON v2024.07 (https://gitlab.dkrz.de/icon/icon-model/). The model is run here in climate limited-area mode (ICON-CLM, Pham et al., 2021), using resources provided by the CLM-Community (www.clm-community.eu), such as ERA5 boundary forcing files and the SPICE environment (v2.2, Rockel and Geyer, 2023). The ICON namelist parameter settings,

which mostly correspond to the defaults in ICON-CLM and those used for operational forecast by the German weather service (DWD), are provided in the supplementary data (Landwehrs et al., 2025). The only modification implemented in the ICON





model code concerns an adjusted heat capacity in the bulk-thermodynamic sea ice model to mimic snow on ice (Littmann, 2024, Sect. 4.2). This ensures quicker and more realistic synoptic variability of near-surface air temperature e.g. in response to WAIs.

ICON is run here in a nested set-up, in which the interior domain covers a region encompassing Svalbard and MOSAiC at ∼2.5 km horizontal resolution (Fig. 1). This interior domain is nested into two larger domains with ∼5 km and ∼10 km resolution, respectively, providing a stepwise transition from the ∼30 km resolution ERA5 reanalysis data (Hersbach et al., 2020). The ERA5 data serve as 3 hourly updated boundary conditions for the regional model at the lower boundary (sea surface temperature and sea ice fraction) as well as the lateral and upper boundaries of the outer model domain. The interior

model domain has 64 vertical levels between 10 m and 17 300 m, of which 12 are below 1000 m. Two ICON model simulations were run for April 2020 to assess effects of Svalbard's topography: the control run $\mathtt{ICON_{ctrl}}$ and an experiment $\mathtt{ICON_{flat}}$ with flattened topography, in which surface height and roughness were reduced by a factor of 1000 but the land-sea mask remained unchanged. Both simulations were initialized on 30 March 2020 and then run continuously for the entirety of April 2020.

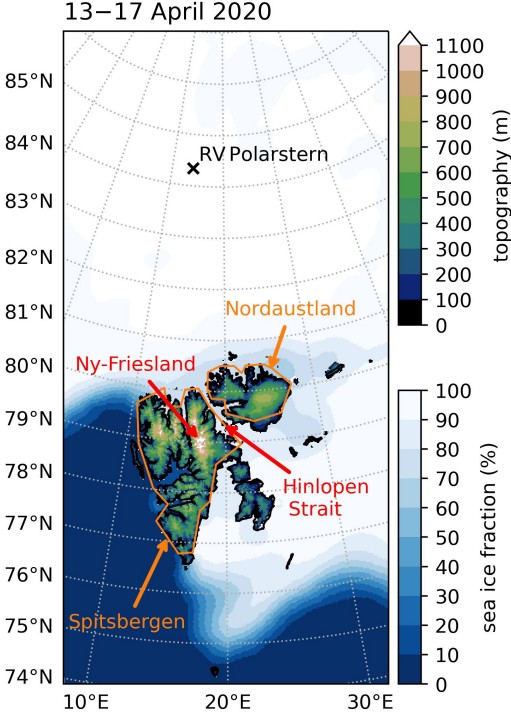

**Figure 1.** Geographical setting of Svalbard. The map section corresponds to the nested interior ICON model domain with 2.5 km horizontal grid resolution. The figure shows the topography and mean sea ice concentration during 13–17 April 2020 at this resolution, as well as relevant topographic features and the position of RV Polarstern.





## 2.2 ICON climate simulations for 2000–2022

To generalize the April 2020 WAI case, we analyze springtime northward moist air advection events over Svalbard using an ICON climate simulation ($ICON_{clim}$), run continuously from 1998 to 2022 over a pan-Arctic domain at 0.1° (∼11 km) resolution. The regional model domain follows the current definition of the Arctic CORDEX domain (https://cordex.org/domains/region-11-arctic/). The general model set-up is similar to that of $ICON_{ctrl}$ and $ICON_{flat}$, with ICON-CLM forced by 3-hourly ERA5 boundary data. However, this simulation employs a slightly older ICON version (v2.6.6, released in January 2023)

and, due to the coarser resolution used for the substantially larger domain, no internal nesting was required. To constrain the large-scale circulation toward ERA5, upper-level grid-point nudging is applied (on wind speed, air temperature and pressure), with nudging coefficients increasing quadratically from 5000 m above sea level upward.

From $ICON_{clim}$, certain time steps were selected to assess northward WAIs over Svalbard in March-April-May (MAM: spring). For this, the hourly column-integrated water vapor transport (IVT) in $ICON_{clim}$ as well as the mean wind speed and

direction at 850 hPa in ERA5 were averaged over a circular region centered on Svalbard (240 km radius around 78.65° N, 18.5° E; see Sect. 3.2.1). ERA5 was used for the 850 hPa wind to filter time steps, as pressure-level data from $ICON_{clim}$ is only available at 6 hourly frequency. Only time steps with a mean wind direction between 165° and 195° (southerly wind) were included. Additionally, different thresholds of the calculated mean IVT and wind speed are applied in Sect. 3.2.1 to further filter time steps with weaker or stronger moisture transport or wind over Svalbard. For the selected timesteps, averages of

different model output variables were computed, using the nearest timesteps of 1 hourly data (IVT, wind speed at 10 m, surface downwelling longwave and shortwave, cloud liquid water path) and 6 hourly data (wind speed, air temperature and relative humidity at 925 hPa, low-level cloud cover).

Furthermore, we diagnose a simple static stability index defined as the potential temperature difference between 600 hPa and 925 hPa within a box south of Svalbard (Sect. 3.2.2).

## 2.3 Lagrangian trajectories for the April 2020 WAI case study

To understand dynamic and thermodynamic effects of Svalbard's topography on warm and moist air masses, this study combines spatially fixed data (Eulerian view) from ICON with a Lagrangian trajectory framework. For air parcels arriving at the MOSAiC site between 14 and 17 April 2020 00 UTC, we compute 36 hour kinematic back-trajectories by running the Lagrangian Analysis Tool (LAGRANTO; Sprenger and Wernli, 2015) on the $ICON_{ctrl}$ model output. Trajectories are initialized

every full hour, horizontally on an equidistant 0.5 km × 0.5 km grid within a circle of 20 km radius centered on the position of RV Polarstern and vertically every 20 hPa between 1000 hPa and 500 hPa (corresponding to heights between ∼13 m and ∼5180 m).

Along each trajectory, the maximum height of topography seen by the air parcel was determined, as well as the difference between the height of the trajectory at the MOSAiC site and its maximum height over topography. These values were then

averaged over all trajectories arriving at the same time and vertical level. To identify trajectories with pronounced warming





or wind speed changes, we computed the differences between temperature and wind speed at the MOSAiC site and the mean values south of Svalbard at 76–77° N.

## 3    Results & Discussion

### 3.1    Topographic effects of Svalbard in the April 2020 WAI

#### 140    3.1.1    Warm and moist air intrusions during April 2020

The MOSAiC expedition, with RV Polarstern near ∼84.3° N, captured two strong warm air intrusion events in mid-April 2020 marked by near-surface temperature peaks on 16 and 19 April. We focus on the first of the two events, which occurred under southerly large-scale flow and a ∼30 K warming from 13 to 16 April, raising near-surface temperatures close to the melting point, together with strongly increased column-integrated water vapor (IWV) and surface downwelling longwave (LWD; Fig. 2).





**Figure 2.** Meteorological conditions at the MOSAiC site during 13–17 April 2020, based on observations and two ICON simulations. Observational data are hourly mean data from the MOSAiC meteorological tower (Cox et al., 2023), except precipitation (KAZR radar 170 m AGL snowfall rate, Matrosov et al., 2022). Differences between ICON$_{ctrl}$ and ICON$_{flat}$ are shown in gray (right hand y-axes). The vertical dashed lines mark the two selected time periods, 14 April 12–24 UTC and 16 April 12–18 UTC, during which the air masses arriving at the MOSAiC site had previously crossed high terrain and undergone subsidence (see Sect. 3.1.1).

During this event, warm and moist air masses were advected northward and partly crossed Svalbard toward the MOSAiC site, as also noted by Svensson et al. (2023). Our trajectory analysis confirms that many air parcels arriving at MOSAiC between 14 and 16 April passed directly over Svalbard's topography (Fig. 3). During the second half of 14 April and the afternoon of 16 April, air parcels initialized at relatively low altitudes ($\lesssim 2$ km) encountered particularly high maximum topography and underwent pronounced subsidence along their trajectories. During these time intervals, we expect the strongest evidence of topographic effects at the MOSAiC site. To characterize them, we selected two representative periods: 14 April 12–24 UTC and 16 April 12–18 UTC, which capture the mean meteorological conditions and topographic influence both at the northern margin of Svalbard and at MOSAiC from an Eulerian perspective. However, the considered air masses typically require ∼6–12 hours





to move between these two areas (Sections 3.1.2–3.1.3), indicating that the topographic effects are not synchronous across the

domain. The topographic forcing was created slightly further upstream when the same air masses crossed Svalbard. To account

for this lag, Figure 4 depicts the atmospheric flow over Svalbard during 14 April 00–12 UTC and 16 April 00–06 UTC, 12

hours before the air arrived at the MOSAiC site during 14 April 12–24 UTC and 16 April 12–18 UTC (Fig. 5).

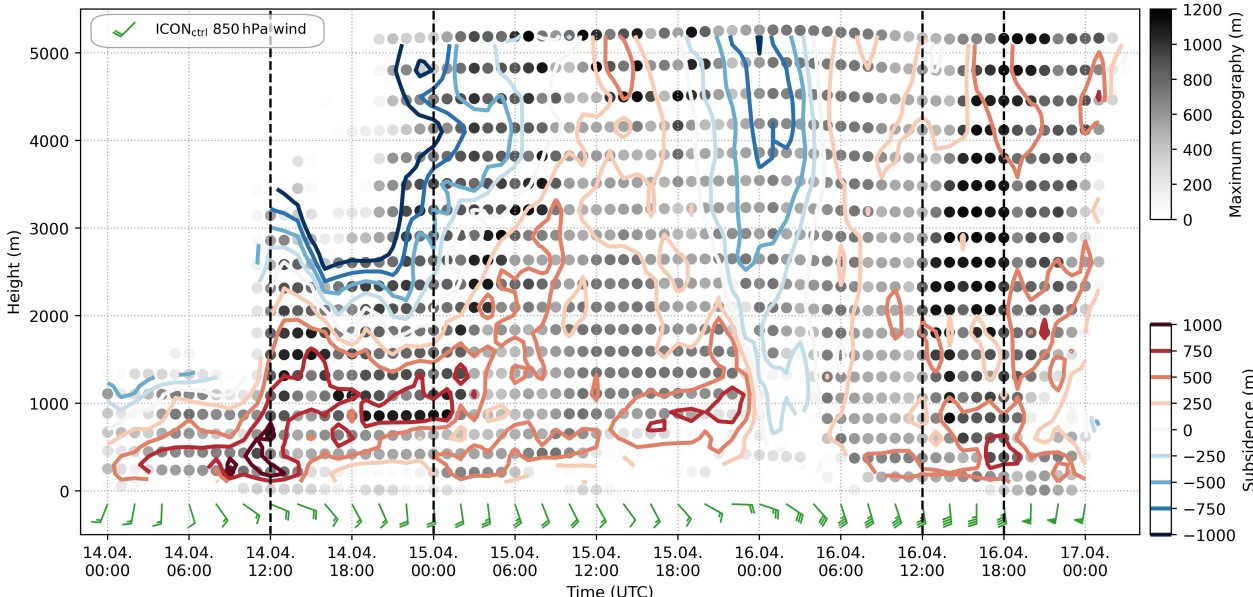

**Figure 3.** Maximum topography and subsidence for trajectories arriving at the MOSAiC site during 14–16 April 2020. The values are averages over trajectories initialized at the same time and height level. Contour lines indicate the vertical height difference between the maximum altitude over land (Svalbard) and the arrival height at the MOSAiC site. Red (blue) contours represent descending (ascending) air parcels relative to their peak height over land. The vertical dashed lines mark the two selected time periods, 14 April 12–24 UTC and 16 April 12–18 UTC.

### 3.1.2   Topographic effects on wind

On 14 April, which marks the initial phase of the warming event at the MOSAiC site (Fig. 2a), the wind at 850 hPa shows

a southerly flow over Svalbard to MOSAiC, while the near surface winds are more south-easterly (Figs. 4a–c, 5a–c; also see

Fig. 2c). This clockwise veering of wind with height is characteristic of warm air advection within the warm sector of a cyclone

to the southwest of Svalbard. Consistent with this, trajectories reaching MOSAiC above 2 km height between 14 April 12 UTC

and 15 April 6 UTC show an ascent over colder air (Fig. 3). Near the surface, orographic forcing produces fast downslope

winds on the northern slopes of Nordaustland and a gap flow through the Hinlopen Strait (Figs. 4a, 5a). An accelerated flow

with increased wind speeds around the topographic barrier is observed both east and west of Svalbard (Figs. 4a–c, 5a–c). The

highest wind speeds are found at the 925 hPa level and exceed $25 \, \mathrm{m \, s^{-1}}$. Between the accelerated flow around Svalbard, wind

speeds at the three depicted height levels are reduced in a wake zone directly north of Svalbard. Compared to the $\mathrm{ICON_{flat}}$




experiment, wind speeds are reduced by more than $5\,\mathrm{m\,s^{-1}}$ even beyond 84° N, while the wind west of Svalbard is accelerated by a similar amount as the flow is diverted by the topographic barrier (Figs. 4a–c, 5a–c).



**Figure 4.** Wind speed and direction in $\texttt{ICON}_{\texttt{ctrl}}$ averaged over 14 April 2020 00–12 UTC (a–c, g) and 16 April 2020 00–06 UTC (d–f, h). The maps show wind speed (WS, shading) and horizontal flow (streamlines) at 10 m, 925 hPa and 850 hPa height. Blue and red contours in all panels indicate the wind speed difference $\Delta\mathrm{WS}_{\mathrm{ctrl-flat}}$ (i.e., $\texttt{ICON}_{\texttt{ctrl}}$ minus $\texttt{ICON}_{\texttt{flat}}$), smoothed with a Gaussian filter. Blue (red) contours indicate a decrease (increase) in wind speed due to Svalbard's topography. The black cross marks the position of MOSAiC. The thick gray contour outlines the 70 % sea ice fraction. The cross-sections (g, h) along the dashed black line in (c, f) show the wind speed on the same color scale as well as potential temperature $\theta$ (dashed black contours) and turbulent kinetic energy (TKE, pink contour).



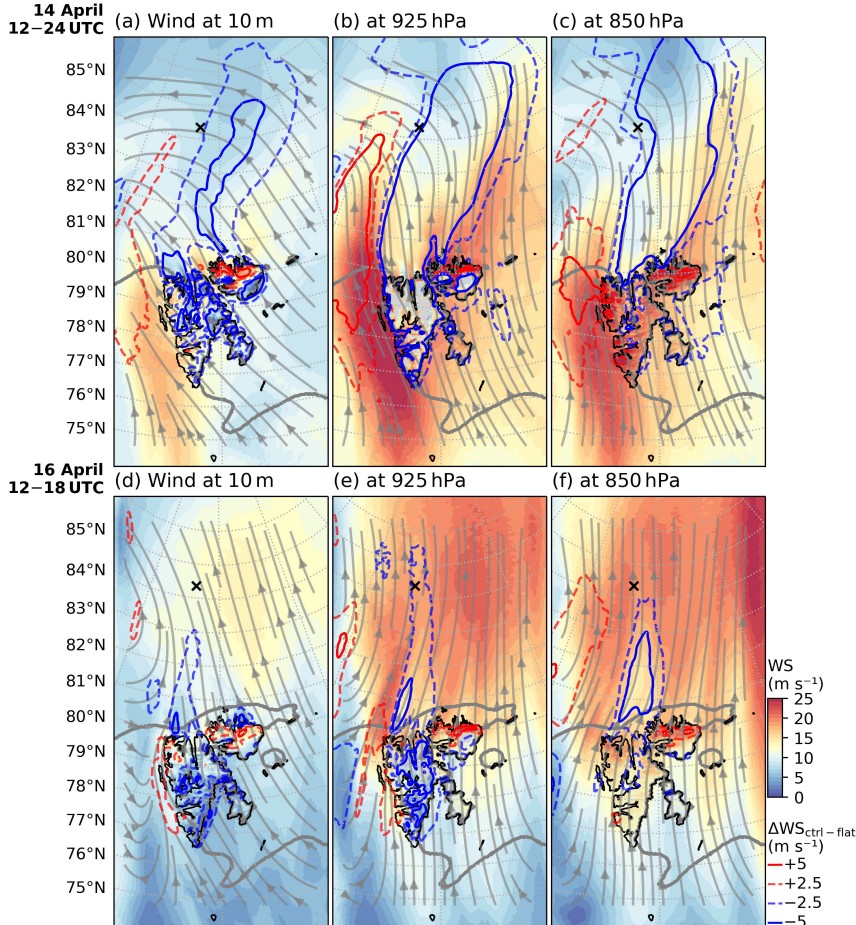

**Figure 5.** Wind speed (WS, shading) and horizontal flow (streamlines) in ICON$_{\text{ctrl}}$ averaged over 14 April 2020 12–24 UTC (a–c) and 16 April 2020 12–18 UTC (d–f) at 10 m (a,d), 925 hPa (b,e) and 850 hPa (c,f) height. Blue and red contours show the wind speed difference $\Delta$WS$_{\text{ctrl−flat}}$, smoothed with a Gaussian filter. Blue (red) contours indicate a decrease (increase) in wind speed due to Svalbard's topography. The black cross marks the position of MOSAiC. The thick gray contour outlines the 70 % sea ice fraction.

The wind speeds modeled by ICON$_{\text{ctrl}}$ at the MOSAiC site are close to the observations, both near the surface and throughout the air column (Figs. 2c, A1b). In contrast, ICON$_{\text{flat}}$ produces wind speeds at the MOSAiC site which are on the order of ~5 m s$^{-1}$ higher than in ICON$_{\text{ctrl}}$ during approximately 14 April 6 UTC to 15 April 6 UTC (Figs. 2c, 6a). The close match between ICON$_{\text{ctrl}}$ and MOSAiC observations confirms that Svalbard's topographic influence on wind speeds in the WAI flow extends downstream and is felt at the MOSAiC site.



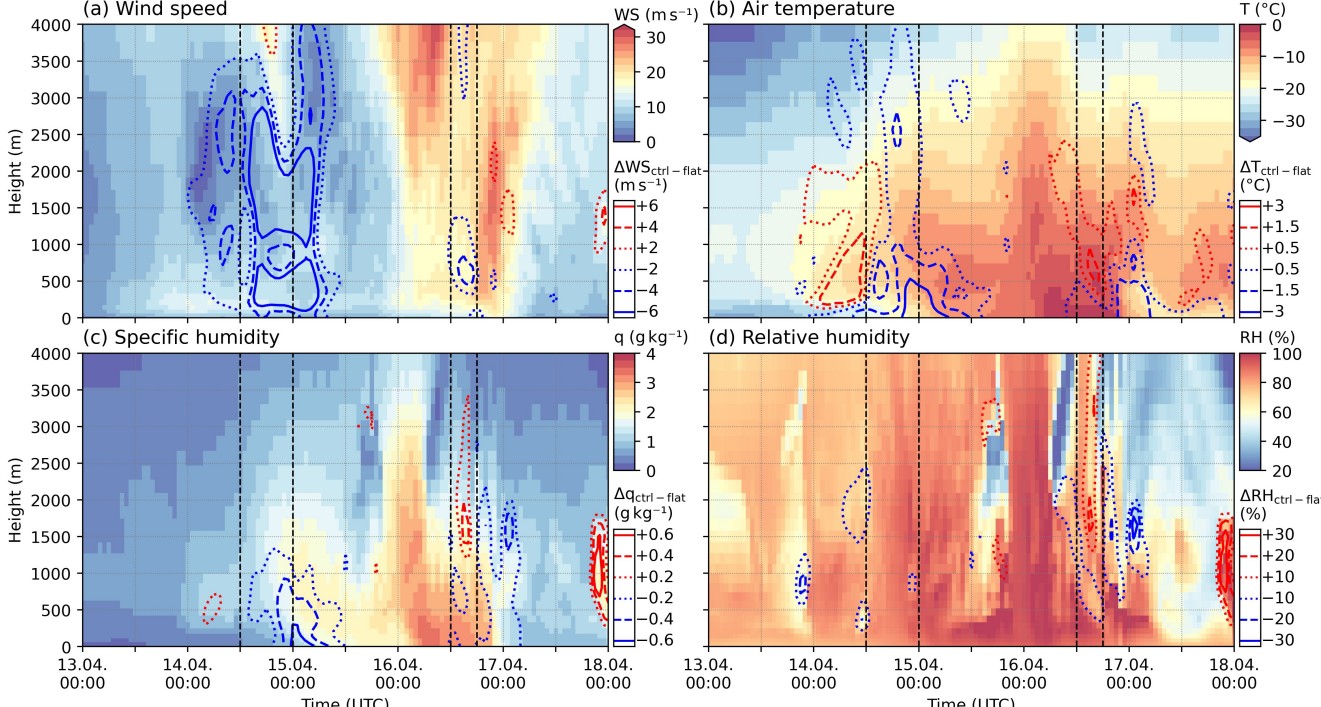

**Figure 6.** Vertical profiles of wind speed (a), air temperature (b), specific humidity (c) and relative humidity (d) at the MOSAiC site during 13–17 April 2020 in ICON$_{ctrl}$ (shading). Contour lines indicate differences ICON$_{ctrl}$ minus ICON$_{flat}$, smoothed with a Gaussian filter. The vertical dashed lines mark two selected time periods, as explained in Fig. 2.

Evaluating the air parcel trajectories initialized at the MOSAiC site on 14 April 21–23 UTC, we find that a large portion of the trajectories exhibits a wind speed reduction of more than $10\,\mathrm{m\,s^{-1}}$ between the windward side of Svalbard (average between 76–77° N) and at the MOSAiC site (Fig. 7). Especially for initial heights of 800–2100 m, more than two thirds of the trajectories experience a slowdown of this magnitude and these air masses cross over the western and central part of Svalbard. When following the decelerated trajectories $\mathrm{Traj}_{\Delta WS<10\,\mathrm{m\,s^{-1}}}$ and particularly the one with the largest wind speed decrease $\mathrm{Traj}_{\Delta WSmax}$ (Fig. 7, red line), we see that many of them are at $\sim$1000–1500 m height on the upwind side of western Spitsbergen at $\sim$77–78° N. In line with Fig. 5, this is where the wind speeds are highest. The trajectories then ascend by $\sim$500 m and pass over Svalbard approximately 12 hours before reaching MOSAiC (Fig. 7c, black markers). Around Svalbard, high turbulent kinetic energy (TKE) throughout the air column indicates perturbation of the flow (Fig. 7a, pink contour). The isentropes and trajectories show a drawdown around $\sim$79–80° N, followed by a hydraulic jump at $\sim$80° N and the vertical decompression and deepening of the flow indicated by the vertical spreading of the isentropes and the trajectories. These processes contribute to the clear reduction of wind speed north of 80° N and especially at 500–3000 m height around 82° N. This is about 6 hours before the air masses reach MOSAiC (Fig. 7c, black markers) where they still carry the reduced wind speeds. The high wind speeds found at 80–81° N below 500 m along the selected trajectory $\mathrm{Traj}_{\Delta WSmax}$ (Fig. 7a) result from the gap



flow of the south-easterly near-surface flow through the Hinlopen Straight (Fig. 4a). Without this channel, reduced downstream
wind speeds could be expected also in this section along the trajectory. However, air masses arriving at the MOSAiC site below
400 m height on 14 April around 21–23 UTC actually travel east of Svalbard and are less directly affected by the topography
(Fig. 7b, and Figs. 5a, 6a).

**Figure 7.** Air mass trajectories arriving at the MOSAiC site on 14 April 2020 21–23 UTC. $\Delta$WS denotes the wind speed change along
trajectories relative to 76–77° N. Red lines in (a)–(b) highlight the trajectory with the largest $\Delta$WS at the MOSAiC site. Panel (a) shows
its vertical profiles of WS (shading), potential temperature $\theta$ (dashed black contours) and turbulent kinetic energy (pink contour). In (b), all
trajectories with $\Delta$WS $< -10$ m s$^{-1}$ at the MOSAiC site are color-highlighted. Panel (c) shows their mean height (gray lines) and $\Delta$WS (dot
color), averaged in latitudinal bins for trajectories initialized at the same height. The text on the right side indicates the number fraction of
this subset of trajectories as well as their averaged $\Delta$WS at the MOSAiC site. The maximum topography along the trajectories is indicated
in gray. Black markers indicate air parcel positions 6 and 12 hours back in time from initialization.





As discussed above, the topographic effects on the wind are very pronounced during 14 April 2020. This is in contrast to 16
April, the second time period with southerly flow from Svalbard towards MOSAiC. During that time, a narrower wake zone ap-
pears north of Spitsbergen (Figs. 4d–f, 5d–f) which reaches up to the MOSAiC site where reduced wind speeds can be observed
in ICON$_\mathrm{ctrl}$ vs. ICON$_\mathrm{flat}$ (Figs. 2a, 6a). However, these effects remain weaker than on 14 April. Particularly, there are few tra-
jectories with a notable wind speed reduction between 76–77° N and the MOSAiC site, and none with $\Delta\mathrm{WS} < -10\,\mathrm{m\,s^{-1}}$ that
would be comparable to those shown in Fig. 7 for 14 April 21–23 UTC.

Why do these two cases differ? Of course, the different synoptic situation with wind speed maxima to the west and to the
east of Svalbard during 14 and 16 April (Fig. 5), respectively, will interact differently with the complex terrain. Nevertheless,
the flow direction over Svalbard at different height levels is comparable between the two time windows (Fig. 5). We find that
part of the different behavior can be explained by a different vertical stability. In the south–north cross-section across Svalbard
towards the MOSAiC site (Fig. 4g–h), the impinging flow on 14 April is capped upstream by more stably stratified air masses,
especially in a layer at ∼2 km that marks the top of the moist air mass. This is indicated by a larger potential temperature
gradient with denser isentropes than on 16 April (also resulting in a higher $N$ and $\hat{h}$, and lower $Fr$; Fig. A2). This stable air
is more resistant against vertical displacement and flow-over the topographic barrier, and therefore promotes partial blocking
and flow-around on 14 April. Consistent with this, we see stronger barrier winds and a stronger gap flow through the Hinlopen
strait on 14 April, as well as streamlines that curve more strongly around the topographic barrier (Fig. 4). This is apparent in
the 925 hPa wind, but also in the 10 m wind around Nordaustland. The topography also causes pronounced upstream blocking
because wind speeds south-west of Svalbard are reduced in ICON$_\mathrm{ctrl}$ compared to ICON$_\mathrm{flat}$. Below the stable air mass the
isentropes are compressed and accelerated winds occur over Svalbard (Fig. 4c,g). Where isentropes exhibit sharp vertical
displacements, TKE is high, indicating wave steepening and turbulent mixing (Fig. 4g). Downstream, the flow expands and
together with momentum loss to turbulence and mountain-wave drag forms a broad lee wake with reduced winds.

In contrast, on 16 April the upstream stratification aloft is weaker (less dense isentropes, lower $N$ and $\hat{h}$, higher $Fr$), which
facilitates a more linear flow over the barrier that remains less perturbed and more laminar (Fig. 9). The isentropes also show
sharp displacements, which however appear to be largely reversible as they are not associated to similarly high TKE (Fig. 4h).
Compression of the isentropes and acceleration over the mountains crests and lee-side wind speed reduction comparable to the
14 April case are not apparent. Still, there is a downslope wind over the lee side of Nordaustland (Fig. 4f).

### 3.1.3 Topographic effects on air temperature

We find that near-surface air temperatures are on the order of 1–4 K lower in ICON$_\mathrm{ctrl}$ vs. ICON$_\mathrm{flat}$ downstream of Svalbard,
except close to the coast where föhn-induced warming emerges (Fig. 8a). At the MOSAiC site, air temperatures in ICON$_\mathrm{ctrl}$ are
relatively close to the observations near the surface and throughout the air column (Figs. 2a, 6b, A1a–b). However, ICON$_\mathrm{ctrl}$
is consistently colder than ICON$_\mathrm{flat}$ at the MOSAiC site during that time period, with temperature differences decreasing
with height, from up to 8 K near the surface to ∼3 K in the lower ∼700 m (Fig. 6b). This reflects the described blocking and
deceleration of the WAI flow by the topography, which leads to a delayed arrival of the warm and moist air at the MOSAiC
site.





On the afternoon of 16 April, $T_{2m}$ is slightly lower in $\texttt{ICON}_{\texttt{ctrl}}$ vs. $\texttt{ICON}_{\texttt{flat}}$, suggesting a similar topographic cooling effect on near-surface temperatures at the MOSAiC site (Figs. 2a, 6b, 8e), where partial blocking of the WAI flow promotes the inflow of colder air from the west (Figs. 5d, 8e). However, this effect remains smaller than the cooling peak around midnight of 14

April. One factor is that on the afternoon of 16 April, downward shortwave radiation (SWD) is enhanced by about $>30\,\mathrm{W\,m^{-2}}$ north of Svalbard and at the MOSAiC site due to reduced lee-side cloud cover (Figs. 2f, 8h). This compensates at least partly for the cooling induced by the blocking of warm air. Downward longwave radiation is reduced by a similar amount in $\texttt{ICON}_{\texttt{ctrl}}$ vs. $\texttt{ICON}_{\texttt{flat}}$, but the enhanced SWD has a smaller net effect on the surface radiation balance due to the high albedo of sea ice. The shortwave warming contribution is absent during the nighttime period on 14–15 April (Fig. 2) which highlights that

Svalbard's effects on surface temperature and energy balance in WAIs over sea ice are also modulated by sunlight conditions, depending on the season and time of day.





**Figure 8.** Air temperature at 2 m (a,e), air temperature at 925 hPa (b,f), and surface downwelling longwave (c,g) and shortwave radiation (d,h) in ICON$_\texttt{ctrl}$ averaged over 14 April 2020 12–24 UTC (a–d) and 16 April 2020 12–18 UTC (e–h). Contours indicate the difference ICON$_\texttt{ctrl}$ minus ICON$_\texttt{flat}$, smoothed with a Gaussian filter. The black cross marks the position of MOSAiC. The thick gray contour outlines the 70 % sea ice fraction.

As discussed, Svalbard's topography causes reduced downstream air temperatures especially around 14–15 April and to a lesser degree on 16 April near the surface. On the other hand, also föhn warming effects are noticeable on the lee side of Svalbard on 14 April 12–24 UTC (Fig. 8a–b). On 16 April, these warming effects stand out more clearly north of Svalbard and up to MOSAiC. In the Eulerian view this is apparent as a ca. +1.5 K air temperature difference in ICON$_\texttt{ctrl}$ vs. ICON$_\texttt{flat}$
at ~300–1200 m and 925 hPa height (Figs. 6b, 8f, A3c). However, the Lagrangian perspective on the air parcel trajectories in Fig. 9 provides the clearest picture of föhn effects. All trajectories arriving at the MOSAiC site on 16 April 15–17 UTC travel from the south over some part of the archipelago (Fig. 9b). A large subset Traj$_{\Delta T>1\,K}$ of them shows a warming $\Delta T > 1$ K





between the windward side of Svalbard (76–77° N) and the MOSAiC site. These air parcels passed over Spitsbergen, about 12
hours before reaching MOSAiC in the case of trajectories below ~2500 m (Fig. 9d). They then experience a drawdown on the
order of 500 m, mostly between 79 and 80° N, as indicated by the median height of the selected trajectories (gray lines) and
by the isentropes (dotted black lines) along the trajectory $\text{Traj}_{\Delta \text{Tmax}}$ with the largest $\Delta$T (red line; Fig. 9a–c). This results in a
mean warming of up to 6 K, which is largest around 80° N at ~300 m for the trajectories intitialized at 383 m height (Fig. 9d).
Subsequently, the trajectories slightly ascend and cool again but still carry the elevated temperatures partly up to the MOSAiC
site. There, the lower-level föhn warming is most evident at 605 m and 835 m height, where $\Delta$T exceeds +1 K for half of the
trajectories and the mean warming is 3.3 K and 2.4 K, respectively. This shows that the positive temperature differences found
in $\text{ICON}_{\text{ctrl}}$ vs. $\text{ICON}_{\text{flat}}$ at this height (Figs. 6b, 8f, A3c) result from föhn effects induced by Svalbard's topography.

The trajectory $\text{Traj}_{\Delta \text{Tmax}}$ (Fig. 9a–c, red line) is drawn down by ~1500 m, warming by more than +10 K and still has
$\Delta \text{T} \approx +8$ K at the MOSAiC site (Fig. 9c). This temperature increase is largely explained by compression and +16 K adiabatic
warming, which is offset by −3 K cooling from latent heat absorption, e.g. by evaporation of cloud droplets and snow. This
suggests that isentropic drawdown (Elvidge and Renfrew, 2016) is the dominant föhn mechanism here. On 14 April, Sval-
bard's topography leads to cooler temperatures at the MOSAiC site, with no net warming effects, as previously discussed.
Nevertheless, nearly half of the air parcel trajectories arriving on 14 April between 21–23 UTC at 800–1100 m show signs of
föhn warming (Fig. A4), similar to 16 April. Topographic warming thus also occurs around the 14 April, especially close to
Svalbard (Fig. 8a–b), but it is less pronounced further north compared to 16 April.

Our results indicate that föhn effects triggered by Svalbard's terrain can extend up to 500 km poleward in both time periods.
Nevertheless, föhn winds only reach the surface over Svalbard or in its immediate vicinity. Further north, including at the
MOSAiC site, föhn signatures are confined to the lower troposphere above the surface.





**Figure 9.** Air mass trajectories arriving at the MOSAiC site on 16 April 2020 15–17 UTC. $\Delta T$ denotes the temperature change along trajectories relative to 76–77° N. Red lines in (a)–(c) highlight the trajectory Traj$_{\Delta Tmax}$ with the largest $\Delta T$ at the MOSAiC site. Panel (a) shows its vertical profiles of relative humidity (shading) and potential temperature $\theta$ (dashed black contours). Panel (c) shows the corresponding air parcel temperature simulated by ICON$_{ctrl}$, along with temperatures diagnosed from adiabatic and latent heat contributions. In (b), all trajectories with $\Delta T > 1$ K at the MOSAiC site are color-highlighted. Panel (d) shows their height (gray lines) and $\Delta T$ (dot color), averaged in latitudinal bins for trajectories initialized at the same height. The text on the right side indicates the number fraction of this subset of trajectories, as well as their averaged $\Delta T$ and height change $\Delta z$ at the MOSAiC site. The maximum topography along the trajectories is indicated in gray. Black markers indicate air parcel positions 6 and 12 hours back in time from initialization.





### 3.1.4 Topographic effects on moisture and clouds

To close the April 2020 case study, we briefly examine topographic effects on humidity and clouds that accompany those on the air temperature (Sect. 3.1.3). During both considered time frames on 14 and 16 April, the southern windward slopes of Svalbard experience high relative humidity, leading to cloud formation and precipitation (Fig. 10). The enhanced cloud cover furthermore contributes to increased LWD and reduced SWD (Fig. 8). In the lee of Svalbard, isentropic drawdown and adiabatic warming reduce the relative humidity, resulting in minimal low-level cloud cover and precipitation (Fig. 10).

On 16 April, these effects on cloud cover are very pronounced north of Svalbard and at the MOSAiC site, where the föhn warming arriving at ∼500–1000 m height also corresponds to a layer with ∼10–15 % lower relative humidity in $\mathrm{ICON_{ctrl-flat}}$ (Figs. 6d, 9a). At the same time, the topography does not clearly reduce the total moisture (IWV) downstream of Svalbard and at the MOSAiC site (Fig. 2b, 6c, 10e). The opposite holds for 14 April, when the topography reduces the moisture ($\Delta$IWV$<$ $-1\,\mathrm{kg\,m^{-2}}$) and precipitation arriving north of Svalbard and at the MOSAiC site (Fig. 2b, 6c, 10a). On the other hand, the

deflected flow around Svalbard is associated with increased IWV on the eastern side and enhanced precipitation on the western side (Fig. 10a, d). Dry föhn air masses also appear north of Svalbard during 14 April (Fig. 10b, A3, A4), but föhn effects are less prominent at the MOSAiC site, as discussed in Sect. 3.1.3.

These cloud and moisture patterns reflect the flow regime contrast established in Sections 3.1.2–3.1.3. On 14 April, the more non-linear regime favors deflection of the moist air, while the more linear regime on 16 April facilitates flow-over with more

pronounced and long-ranging föhn effects.





**Figure 10.** Column-integrated water vapor (a,e), relative humidity at 925 hPa height (b,f), low-level (>800 hPa) cloud cover (c,g) and total precipitation (d,h) in in `ICON_ctrl` averaged over 14 April 2020 12–24 UTC (a–d) and 16 April 2020 12–18 UTC (e–h). Contours indicate the difference `ICON_ctrl` minus `ICON_flat`, smoothed with a Gaussian filter. The black cross marks the position of MOSAiC. The thick gray contour outlines the 70 % sea ice fraction.

## 3.2 From case study to climatology: topographic effects of Svalbard in poleward warm and moist air advection during spring

### 3.2.1 Climatological composites for high and low moisture transport and wind

To generalize the insights gained from the April 2020 case study in a climatological context, we now analyze springtime periods with southerly advection over Svalbard during 2000–2022 in `ICON_clim` (Sect. 2.2). For March-April-May 2000–2022, we identified 737 and 821 hourly time steps where the mean moisture transport ($IVT_{Sv}$) or wind speed ($WS_{850\,hPa,Sv}$) over



Svalbard, respectively, was in the upper quartile and from southerly direction (165–195°). For 380 time steps, these conditions were simultaneously met. Composite mean fields for these subsets are shown in Fig. 11. We refer to the 25–50 %ile and 75–100 %ile of $IVT_{Sv}$ (27.8–48.2 and >79.4 $kg\,m^{-2}\,s^{-1}$) or $WS_{850\,hPa,Sv}$ (5.5–7.8 and >10.8 $m\,s^{-1}$) as "low" and "high" IVT or wind speed. $IVT_{Sv}$ and $WS_{850\,hPa,Sv}$ are correlated and their mean values increase between the 25–50 %ile and 75–100 %ile of the other respective quantity (45.3 to 93.1 $kg\,m^{-2}\,s^{-1}$ and 7.4 to 12.6 $m\,s^{-1}$).

Comparing results for these climatological composites with the topographic effects identified for the April 2020 case (Sect. 3.1), similar key features can be recognized. Near-surface winds are accelerated in the gap flow through Hinlopen Strait and on the lee side slopes of Nordaustland in combination with low-level blocking on its south-western windward slope (Fig. 11a), similar to the situation on 14 April 2020 (Fig. 5a). The lowest wind speeds occur over central Spitsbergen, which has the roughest terrain, in contrast to the smoother surface of Nordaustland and its ice caps. Downstream, to the north and north-east of Spitsbergen, reduced wind speeds at 10 m and 925 hPa height are identified up to at least 82° N (Fig. 11a–b). This stands out against the accelerated flow around Svalbard, with the highest wind speeds on its western flank. Near the surface, the higher wind speeds west of Svalbard also partly result from the lower surface roughness of the open ocean compared to the sea ice east of Svalbard. Comparing time steps with high and low $IVT_{Sv}$ or $WS_{850hPa,Sv}$, wind speeds at 10 m and 925 hPa most strongly increase in the flow around Spitsbergen and Nordaustland (Fig. 11a–b, red contours), but much less over central Spitsbergen and downstream of that. This indicates stronger topographic perturbation and destabilization of the flow at high wind speeds compared to moderate wind speeds, as also discussed for the contrasting flow regimes on 14 and 16 April 2020 (in Sect. 3.1).

In addition to the topographic effects on wind, Fig. 11 illustrates föhn effects resulting in elevated air temperatures and reduced humidity and cloudiness, leading to reduced (increased) downwelling longwave (shortwave) north of Svalbard. As noted for the MOSAiC cases, the föhn signal in the absolute temperatures can be relatively subtle, and for the climatology neither Lagrangian trajectory analyses nor an experiment with flattened topography were feasible. However, air temperatures up to 81–82° N downstream of Svalbard appear ~1–2 K higher than to the east and west at those latitudes (Fig.11e). This area shows the largest temperature increase between high vs. low $IVT_{Sv}$ compared to the surrounding (Fig.11e). This indicates stronger föhn warming in moister poleward advection events, potentially due to the release of more latent heat when moister air ascends on the windward side (Elvidge and Renfrew, 2016; Mattingly et al., 2023). Stronger föhn warming also occurs at higher wind speeds (Fig.11e, red solid contour), possibly also related to higher $IVT_{Sv}$.



**Figure 11.** Atmospheric conditions during southerly warm and moist advection in ICON_clim for March-April-May 2000–2022. The color shading shows averages over time steps with ICON IVT ($IVT_{Sv}$) and ERA5 850 hPa wind speed ($WS_{850hPa,Sv}$) above the upper quartile, both within 165–195° mean direction (solid black lines) in the circle marked in (e). For 6 hourly model output variables (b,c,e,f) and daily sea ice fraction (grey contours for 70 %), the timesteps closest to those filtered timesteps were averaged. The solid contours indicate mean differences between time steps with high and low southerly 850 hPa wind speed (> 75 %ile minus 25–50 %ile). Dashed contours show the same for high and low IVT. The location of MOSAiC on 16 April 2020 is marked by the black cross.

Föhn effects are most evident in terms of reduced relative humidity, low-level cloud cover and cloud liquid water path north of Svalbard compared to the flow around the archipelago, where they are actually increased by the influence of Svalbard (Fig.11 c, d, f). These effects are visible up to 83–84° N, showing that Svalbard's topography can influence the atmosphere several hundred kilometers to the north during strong poleward advection events. At higher wind speeds relative humidity decreases north of Svalbard, and low-level cloud cover decreases by more than 5 % up to 84° N (Fig. 11c, solid blue contour).





In this Eulerian analysis, the geographical extent of topographic effects is determined by how long the persistent southerly
advection over Svalbard lasts, and therefore how far the air can travel within that time with a given wind speed. The Lagrangian
analyses in Sect. 3.1 show that air masses need ∼12 hours to travel over Svalbard to >84° N, and some remote effects are likely
missed in the Eulerian approach. For high vs. low $IVT_{Sv}$, the low-level cloud cover is further reduced up to ∼82° N (Fig. 11c,
dashed blue contour) due to stronger föhn, but this effect does not extend as far north as for high vs. low $WS_{850hPa,Sv}$. This is
probably because more moisture is available in the high $IVT_{Sv}$ cases overall, which increases relative humidity and cloudiness
(Figs. 11c, d, f), offsetting drying locally induced by the topography.

Ultimately, we find that the surface downwelling longwave radiation is reduced by approximately $20\,\mathrm{W\,m^{-2}}$, while the
downwelling shortwave radiation increases by about $40\,\mathrm{W\,m^{-2}}$ on the leeside of Svalbard, relative to the surrounding flow
(Fig. 11g, h). These changes reflect the presence of drier and less cloudy air downstream, in contrast to the more humid condi-
tions associated with the flow around the topography.

**3.2.2    Climatological composites for high and low stability**

The MOSAiC case study highlighted that Svalbard's topographic effects depend on the stability of the impinging flow (Sect. 3.1.2).
To generalize this behavior, we diagnose static stability from the potential temperature difference between 925 hPa and 600 hPa
within a box south of Svalbard (see Fig. 12a). We only consider time steps with southerly flow (direction 165–195°) and wind
speed above the 25%ile in the ERA5 wind at 850 hPa within the circle around Svalbard (see Sect. 3.2.1). Figure 12 shows
composites for the upper ($\Delta\theta_{600-925\,\mathrm{hPa}}$ >19.6 K) and lower (<13.3 K) quartiles of this stability index.

In the high stability composite, Svalbard lies between a high pressure ridge to the east and a low pressure region farther west
toward Greenland (Fig. 12a). These systems are traveling eastward so that the ridge covered Svalbard 24 hours prior (Fig. A5).
The high pressure influence and associated subsidence established strong static stability in the free troposphere. Beneath this
stable cap, the approaching low pressure system feeds the warm and moist flow from the North Atlantic. The clockwise veering
of the wind with height is well expressed over and south of Svalbard (Fig. 12a–c streamlines and geopotential height contours).
This indicates baroclinic conditions, linked to the temperature gradient from the arriving warm air south and west of Svalbard,
as well as vertical shear that can contribute to turbulence.

In the low stability composite, the cyclone has moved farther east so that Svalbard lies on its eastern flank (Fig. 12d), while
the ridge retreats. The low pressure system draws cooler air from east of Greenland into the southerly flow over Svalbard,
and the moist air stream from lower latitudes shifts farther east of Svalbard (Fig. A5). The cyclones causes air to ascend and
creates as less stable stratification of the troposphere over Svalbard. Under these conditions, the impinging flow is vertically
more coherent and mixed and thus less stable around Svalbard. Temperature gradients are also less pronounced, contributing
to a more barotropic flow with weaker vertical veering of the wind (Fig. 12d–f).

The two different cases discussed for MOSAiC in April 2020 (Sect. 3.1.2) align well with these regimes. In the high stability
composite and on 14 April 2020, the high pressure influence creates a stable cap and strong winds and warm and moist air
advection are focused on Svalbard's western side (Figs. 4a–c, g, 8a–d, 10a–d, 12a–c). The higher stability aloft favors a flow-
around response with stronger barrier and gap flows and a more pronounced lee wake (Fig. 12). Stronger baroclinity and





vertical shear potentially enhance turbulence and perturbation of the flow over Svalbard. In the low stability composite and in the 16 April 2020 case, stronger winds and warm and moisture transport occur on Svalbard's eastern side, while they are low to its west where the center of the low pressure system is (Figs. 4d–f, h, 8e–h, 10e–h, 12d–f). The weaker tropospheric stability promotes a flow-over regime with overall less pronounced topographic winds. Under the more barotropic conditions, turbulence from vertical shear is also less likely.

In summary, the stability composites and case studies suggest a recurring synoptic sequence: As a cyclone approaches from the west, Svalbard first experiences southerly flow with relatively high stability, favoring flow-around. One or two days later, the eastward moving cyclone brings less stable air which favors a flow-over regime with less perturbation of the flow.





**Figure 12.** Circulation and wind fields over Svalbard for high (a–c) and low (d–f) stability of the impinging southerly flow. The upper and lower quartile of $\Delta\theta_{600-925\,\mathrm{hPa}}$ in the black box south of Svalbard (a,d) were used to define the high and low stability composites, including only time steps with ERA5 850 hPa wind in the circle around Svalbard (see Fig. 11) within 165–195° N mean direction and wind speed above the 25%ile. Shown are the mean sea level pressure together with 850 hPa wind streamlines and 700 hPa geopotential height (a,d), as well as streamlines and wind speed for the wind at 10 m (b,e) and 925 hPa (c,f) height. The black cross marks the position of MOSAiC. The thick gray contour outlines the 70 % sea ice fraction.

## 4 Conclusions

This study demonstrates that Svalbard's complex topography modulates warm and moist air mass intrusions (WAIs) entering the central Arctic via the North Atlantic pathway. A strong WAI event in April 2020 is studied using high-resolution ICON model simulations, Lagrangian trajectory and observations from the MOSAiC expedition. In this case, Svalbard produced two distinct responses whose imprint was detectable up to ~500 km downstream and at the MOSAiC site: flow-around with partial





blocking and a lee wake, and flow-over with föhn signatures. Stronger static stability above ~1.5 km on 14 April favored flow-around, resulting in amplified barrier and gap winds and a broad lee wake. Within this wake, Svalbard's topographic effects cause reduced wind speeds (reduced by $>5\,\mathrm{m\,s^{-1}}$), near-surface air temperatures ($>3\,\mathrm{K}$) and column moisture ($>1\,\mathrm{kg\,m^{-2}}$), as the terrain blocked, perturbed and delayed the warm and moist inflow. On 16 April, the impinging flow was less stable,
facilitating the flow-over response with weaker barrier winds and a less pronounced lee wake. A branch of the southerly flow nevertheless experienced strong lee-side drawdown, yielding föhn warming and drying. This effect is strongest at Svalbard's northern shore and reaches down to the surface. However, a föhn warming of 2–3 K in the lower troposphere is still traced downstream to the MOSAiC site. The föhn also causes a long-ranging reduction of the low-level cloud cover (by $>20\,\%$) which leads to reduced (increased) downwelling longwave (shortwave) radiation at the surface ($>20\,\mathrm{W\,m^{-2}}$). While this case
study focused on time periods when MOSAiC was downstream of Svalbard, southwesterly winds on 18–19 April shifted the wake eastwards so that MOSAiC captured the flow-around north of Svalbard (Fig. A6). This highlights that the local impacts of Svalbard's influence on WAIs strongly depend on the wind direction.

Beyond the April 2020 case study, our springtime (MAM) climatological composites for 2000–2022 indicate that similar signals recur frequently during southerly advection over Svalbard, extending several hundred kilometers poleward and
strengthening with higher wind speed and IVT. We also showed that southerly flow over Svalbard is often associated with a low pressure system approaching from the west and a high pressure ridge to the east, and that the vertical stability of the impinging flow reflects the respective influence of these systems. When the influence of the ridge still persists over Svalbard, stability and flow-around are stronger, and the warm and moist air flow is largely channeled on Svalbard's western side in the Fram Strait. One to two days later, as the cyclone reaches Svalbard and shifts the warm and moist air flow east of Svalbard,
ascent ahead of the system lowers static stability and promotes flow-over. The situation on the 14 and 16 April 2020 exemplify these higher and lower stability regimes, respectively.

Our findings suggest that Svalbard's topography influences the regional impacts of WAIs on the surrounding sea ice. The mechanical forcing is altered as southerly winds intensify in barrier winds and gap flows but weaken in the wake zone, especially under flow-around conditions. For example, barrier winds may increase the northward displacement of the sea ice edge by
WAIs which often pass through the Fram Strait, thereby influencing the sea ice variability in this region. Thermodynamically, föhn effects on near-surface temperature in coastal areas and on the surface energy balance over longer distances could modulate sea ice melt and growth. Together, these dynamic and thermodynamic topographic effects likely result in characteristic shifts of the sea ice concentration around and north of Svalbard, including the opening of coastal polynyas. A complementary next step is to quantify how the terrain modulates precipitation and föhn winds on Svalbard, for example, and to assess the
implications for snowpack and glacier mass balance in a warming climate.

*Code and data availability.* ICON model output for the analyzed variables, ICON namelist parameters, and Python scripts to replicate the figures in this paper are available from Landwehrs et al. (2025) (https://doi.org/10.5281/zenodo.17054204). Data from ERA5 (Hersbach et al., 2020) and MOSAiC (Matrosov et al., 2022; Dahlke et al., 2023; Cox et al., 2023) are publicly available.



*Author contributions.* JL, AR, SM, FG and EG conceptualized the study and wrote the manuscript. JL conducted the investigation (incl. ICON
model simulations), formal analysis and visualization. FG contributed to the investigation by running LAGRANTO.

*Competing interests.* The authors declare that they have no competing interests.

*Acknowledgements.* The authors thank Lukas Papritz, Raphael Köhler, Michael Tjernström and Gunnilla Svensson for valuable discussion.
The CLM-Community is thanked for providing ERA5 boundary forcing files for ICON. The artificial intelligence tool ChatGPT was used to
assist the development of Python code for analyses and visualization as well as for revising the manuscript text.

*Financial support.* JL, EG and AR acknowledge the funding from the European Union's Horizon 2020 Research and Innovation Framework
Programme under Grant Agreement no. 101003590 (PolarRES). FG and AR acknowledge the funding by the Deutsche Forschungsgemein-
schaft (DFG, German Research Foundation) - project 268020496 TRR 172, within the Transregional Collaborative Research Center "ArctiC
Amplification: Climate Relevant Atmospheric and SurfaCe Processes, and Feedback Mechanisms (AC)[3]". SM acknowledges the funding
from the Knut and Alice Wallenbergs Stiftelse under Grant no. 2016-0024, and the Swedish Research Council under Grant no. 2022-03052.
This work used resources of the Deutsches Klimarechenzentrum (DKRZ) granted by its Scientific Steering Committee (WLA) under project
ID49 "Modellierung der atmosphaerischen Zirkulation der Arktis".



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



## Appendix A: Appendix

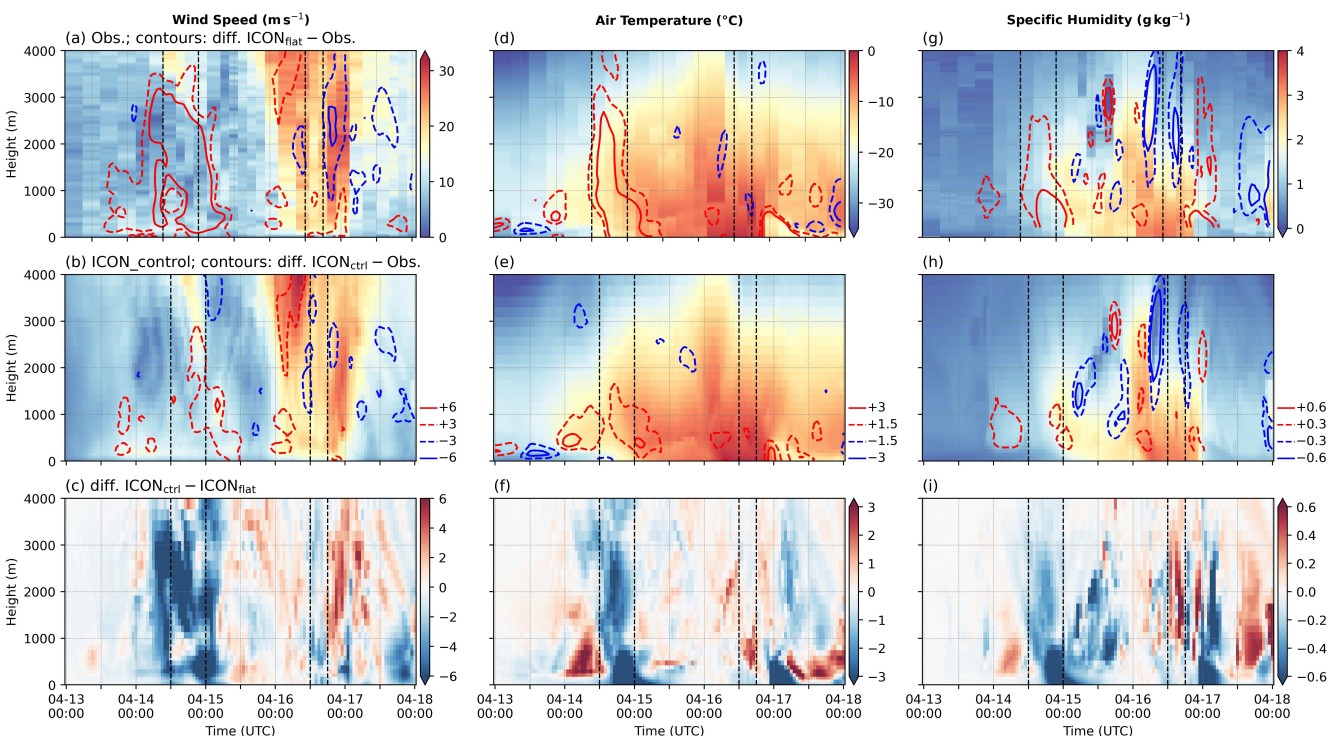

**Figure A1.** Vertical profiles of wind speed (a–c), air temperature (d–e) and specific humidity (g–i) at the MOSAiC site during 13–17 April 2020. The background shading show MOSAiC radiosonde observations (`Obs`) from Dahlke et al. (2023) (top row: a,d,g), output from `ICON_ctrl` (middle row: b,e,h) and the difference `ICON_ctrl−flat` (bottom row: c,f,g). Contour lines indicate differences `ICON_flat − Obs` (top row: a,d,g) and `ICON_ctrl − Obs` (middle row: b,e,h), smoothed with a Gaussian filter. The vertical dashed lines mark two selected time periods, as explained in Fig. 2.



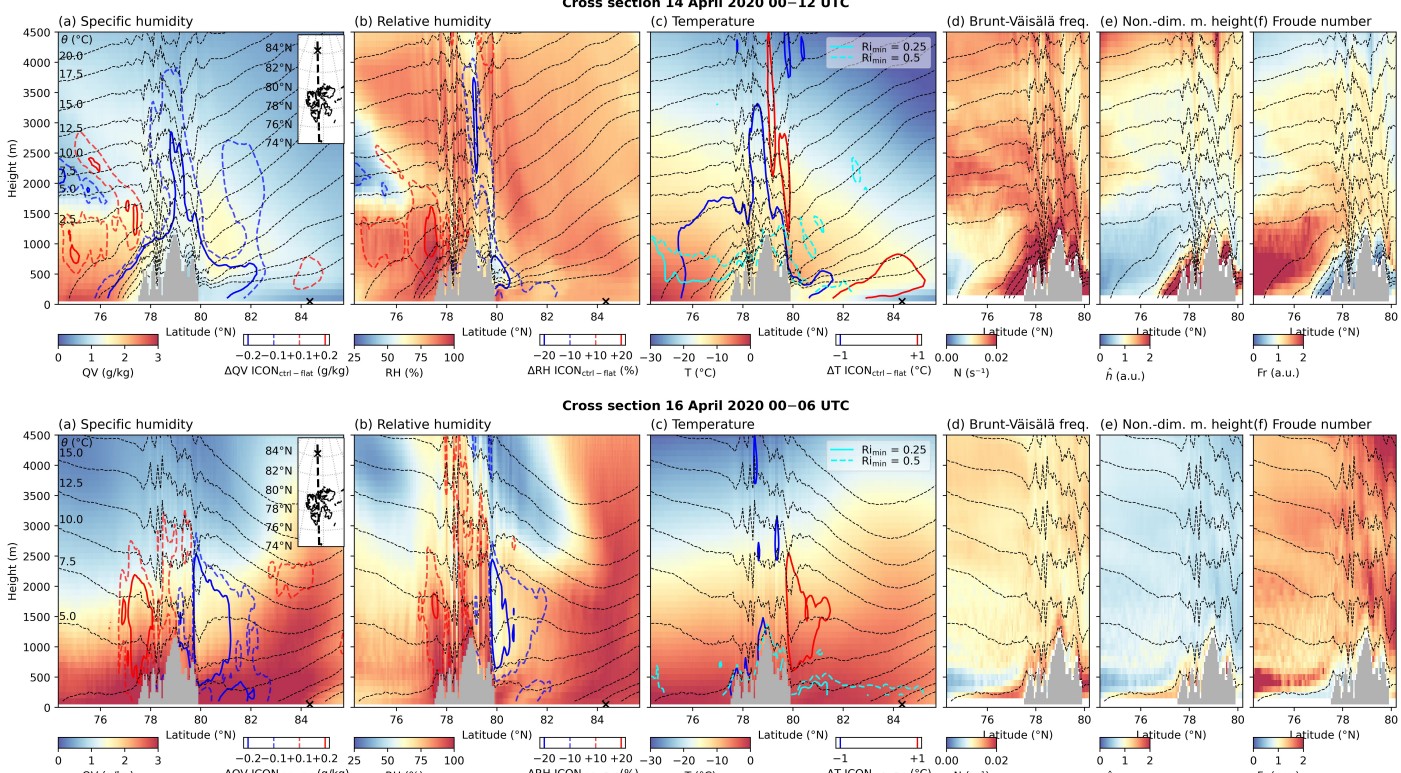

**Figure A2.** Cross section across Svalbard to MOSAiC (dashed black line in inset map) during 14 April 2020 00–12 UTC (top) and 16 April 2020 00–06 UTC (bottom). Shown are time averages of the specific humidity (a), relative humidity (b), air temperature (c), Brunt-Väisälä frequency (d), non-dimensional mountain height (e), Froude number (f). Blue and red contours in all panels indicate the difference $\mathrm{ICON_{ctrl-flat}}$ of the respective quantity, smoothed with a Gaussian filter. Additionally shown are the potential temperature $\theta$ (dashed black contours) and time minimum values of the Richardson number (light blue contours in (c)).



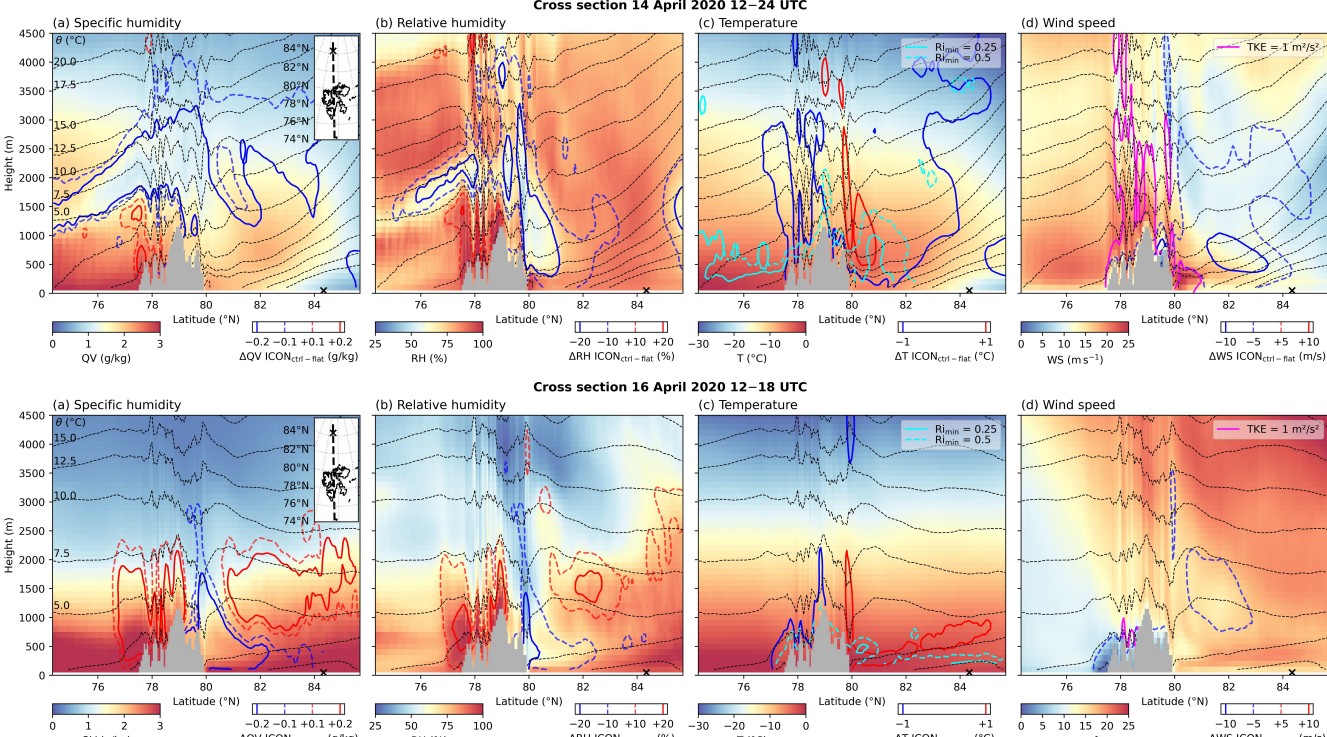

**Figure A3.** Cross section across Svalbard to MOSAiC (dashed black line in inset map) during 14 April 2020 12–24 UTC (top) and 16 April 2020 12–18 UTC (bottom). Shown are time averages of the specific humidity (a), relative humidity (b), air temperature (c) and wind speed (d). Blue and red contours in all panels indicate the difference ICON_ctrl-flat of the respective quantity, smoothed with a Gaussian filter. Additionally shown are the potential temperature $\theta$ (dashed black contours) and time minimum values of the Richardson number (light blue contours in (c)).





**Figure A4.** Same as Fig. 9, showing trajectories with >1 K warming, but for 14 April 2020 21–23 UTC.



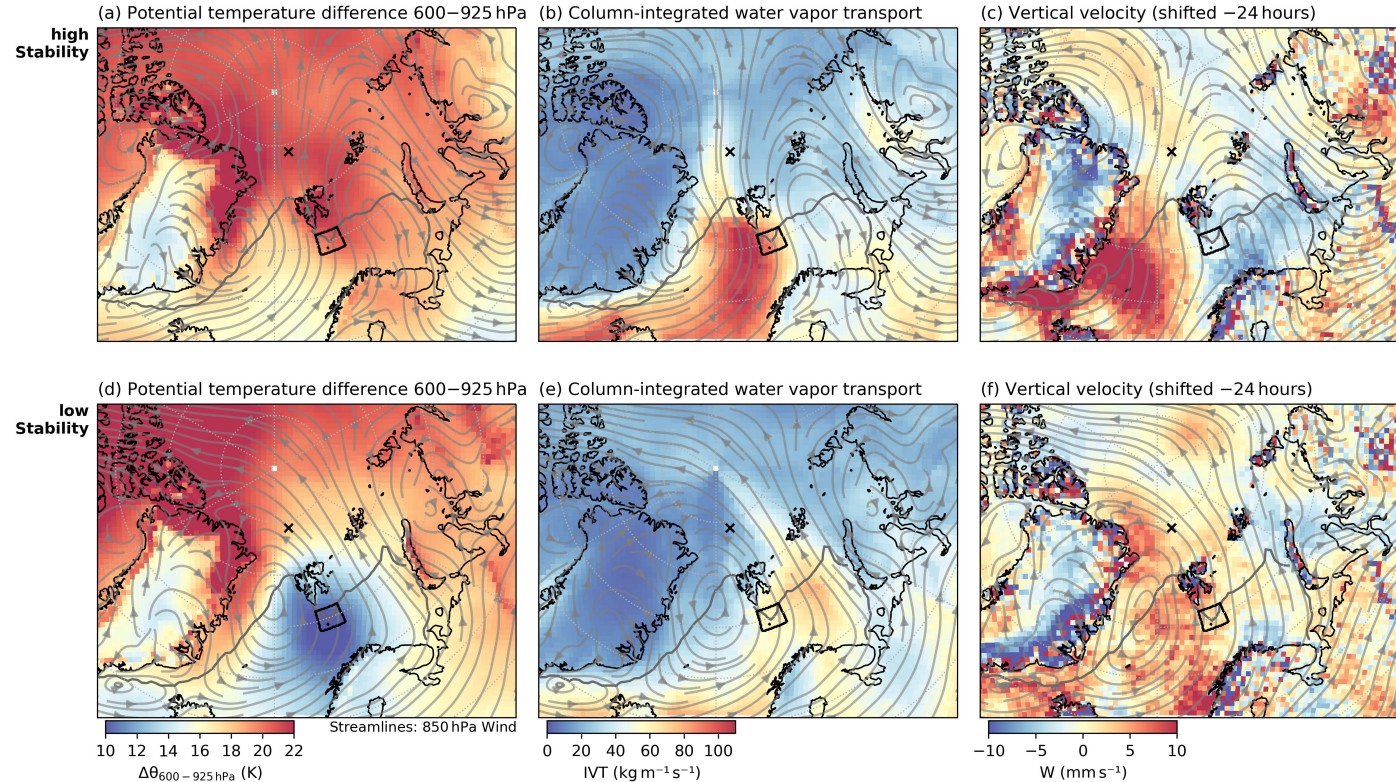

**Figure A5.** Climatological composites of meteorological conditions around Svalbard for high (a–c) and low (d–f) stability of the impinging southerly flow during March-April-May 2000–2022 in ICON$_{\texttt{clim}}$ (compare Fig. 12). Shown are the stability index $\Delta\theta_{600-925\,\text{hPa}}$ (a,d), column-integrated water vapor transport (b,e) and the vertical velocity 24 hours earlier (c,f). The black cross marks the position of MOSAiC. The thick gray contour outlines the 70 % sea ice fraction.



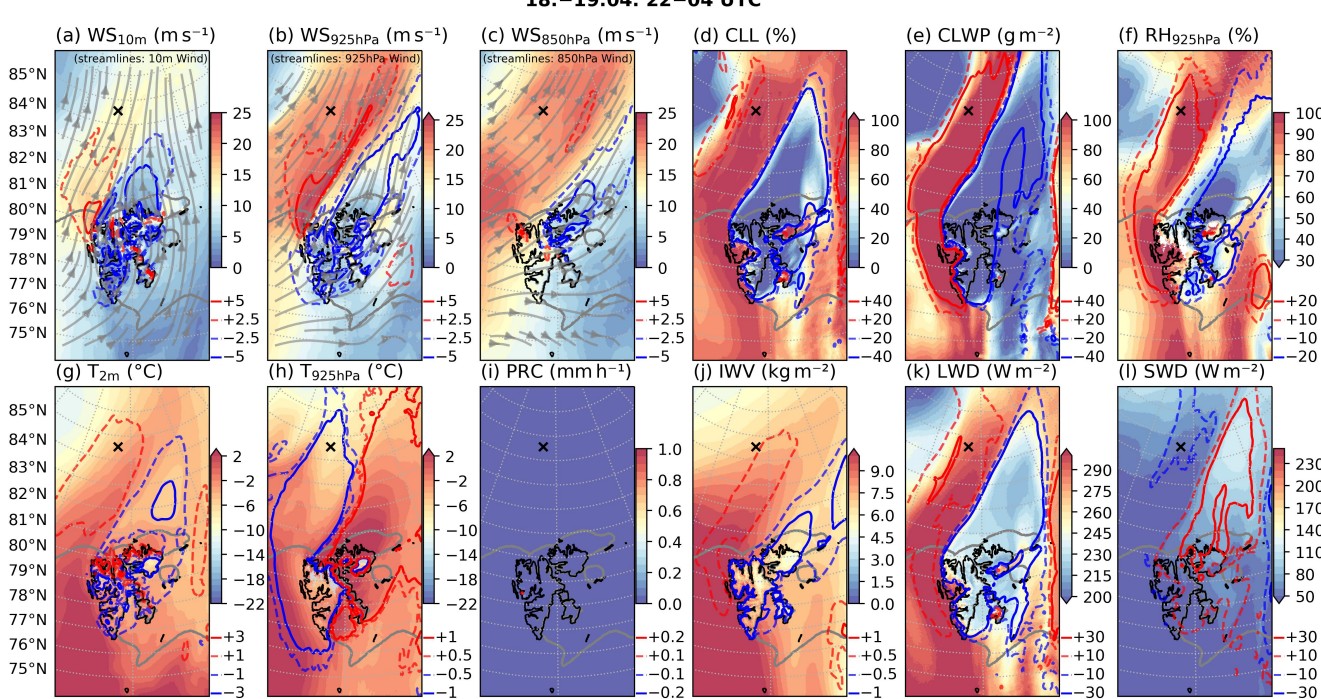

**Figure A6.** ICON_ctrl simulation, averaged over 18–19 April 2020 22–04 UTC. Wind speed at 10 m, 925 hPa and 850 hPa height (a,b,c), low-level cloud cover (d), cloud liquid water path (e), relative humidity at 1058 m above ground model level (f), Temperature at 2 m and 928 hPa (g,h), total precipitation (j), surface downwelling longwave and shortwave (k,l). Radiative fluxes are positive downward. Contours indicate the difference ICON_{ctrl−flat}. Streamlines indicate the wind at 925 hPa, 850 hPa and 10 m height, respectively. The black cross marks the position of MOSAiC. The thick gray contour outlines the 70 % sea ice fraction.