# Peer review of "Topographic Effects of Svalbard on Warm and Moist Air Intrusions into the Central Arctic"

_EGUsphere, 2025_

## Author Response (AR1)

**Author's final response**

We thank the two reviewers for their positive evaluation of our manuscript. We addressed all comments and suggestions and modified the manuscript accordingly.

Font colors:  reviewer comments, author's response, author's changes in the manuscript

**Reviewer 1**

**Review of "Topographic effects of Svalbard on warm and moist air intrusions into the central Arctic" by Landwehrs et al.**

**Citation: https://doi.org/10.5194/egusphere-2025-4535-RC1**

This manuscript presents a detailed model based case study of the impact of Svalbard's topography on a warm air intrusion event in the Arctic, that impacted the MOSAiC expedition in April 2020 and demonstrates the impact that Svalbard's topography has on downstream atmospheric conditions. The case study analysis is thorough and contrasts strong versus weak stability flow over Svalbard. A longer time period climatological analysis is also presented confirming that results seen in the case study occur regularly. The manuscript is well written and the figures clearly display the relevant results. As such I recommend accepting this manuscript with very minor revisions, described below.

 Minor comments

Line 130: It would be useful to state the number of trajectories initialized at each pressure level and the total over all pressure levels.

- We added:

    - „This results in 151320 trajectories, of which 1520–1640 are initialized per hour. Every hour includes 75–82 trajectories on each of the 20 height levels, reflecting small variations in the number of horizontal initialization points."

Figure 3: In the figure caption or text please explain why not all times and heights have plotted values in this figure. Is it because only trajectories that have passed over Svalbard are shown so times and heights without plotted data indicate trajectories that did not pass over Svalbard?

- We added:

    - „Blank areas indicate initialization times and heights where none of the trajectories passed over Svalbard."

Line 196: Fig 2a should be Fig 2b since this sentence is referring to wind speed not temperature.

    - Changed to „Fig. 2c", which is the panel showing wind speed.

Review: "Topographic Effects of Svalbard on Warm and Moist Air Intrusions into the Central Arctic", J. Landwehrs et al., DOI: 10.5194/egusphere-2025-4535

Citation: https://doi.org/10.5194/egusphere-2025-4535-RC2

The manuscript by Landwehrs et al. investigates the impact of Svalbard's up to ~1700 m high topography on warm, moist air masses intruding into the inner Arctic. The authors begin with a detailed case study of an intense mid-April 2020 Warm Air Intrusion (WAI). They combine ICON simulations (basically a "control" vs. "flat Svalbard" sensitivity study) with shipborne observations collected aboard research vessel Polarstern during the MOSAiC expedition. To further investigate the case, a Lagrangian trajectory analysis is conducted and used to complement the Eulerian model evaluations. Finally, the findings of the case study are generalized, using an ICON model climatology extending over ~20 years in time. Through the wealth of methods applied in a meaningful way, the impact of Svalbard's mountain ranges on downstream winds, air temperatures, and the surface energy budget in the central Arctic are clearly unraveled.

Overall, this study is conducted with a sound methodological approach, written well, and offers clear and insightful figures. As I am not aware of any such comprehensive investigation of Svalbard's topographic impact on WAIs existing yet, I think the article will be a very good contribution to the research field.

Below are some minor specific comments I have. All things considered, I recommend accepting the manuscript with minor revisions.

Specific Comments:

- Lines 6-9: Here and throughout the manuscript, you mostly limit your analysis to the effects in the lee of Svalbard (such as the decrease of wind speeds, near surface temperatures etc.). But as you showed in your figures and shortly hinted towards, the effects to the west and east of Svalbard (such as an actual increase in wind speeds) seem just as important and interesting to me. Maybe one or two additional sentences could be added throughout the article to highlight the effects outside the lee more.

- We agree with the reviewer that the effects outside of the lee are important. These aspects are discussed at several occasions in the manuscript (e.g. lines 164–169, 205–209, 274–276, 298–299, 314–316, 366–367, 382–384, 387–390) where we address the flow around Svalbard with accelerated wind speeds, for example. We have now modified the abstract to emphasize these effects more clearly and earlier in the paper:

  ○ *„The response depends on the static stability of the impinging flow: stable conditions favor flow-around response, characterized by accelerated barrier winds along the eastern and western flanks of Svalbard and gap winds through the Hinlopen Strait, together with a broad lee wake north of the archipelago. In this wake, wind speed, near-surface temperature, and column-integrated water vapor are reduced by >5 m s−1, >3 K, and >1 K, respectively."*

- Line 35: There is a comma or semicolon missing before "Elvidge et al.".

- Changed accordingly.

- Lines 41+44: I prefer the term "cloud dissipation" over "cloud clearance".

- Changed accordingly.

- Line 70: please introduce the abbreviation "RV" in "RV Polarstern".

- Changed accordingly.

- Line 82: Why did you limit your climatological analyses to spring? There might be some differences in Svalbard's topographic influence on WAIs between the seasons. Vertical stability and flow regimes most probably differ, so it might have been interesting to see a seasonal analysis. Furthermore, I would be interested to know if the topographic impact is evolving over time, given that the Arctic climate system is rapidly changing. But these points are more food for thought than a criticism of the article.

- We totally agree, but we limited the climatological analysis to spring in order to maintain a clear and coherent narrative centered around the April 2020 MOSAiC case. We added a sentence in the conclusions:

- *„This study focused on the April 2020 MOSAiC WAI case and thus on spring conditions with southerly advection, identifying several factors that modulate Svalbard's topographic effects, including static stability, synoptic-scale circulation patterns, wind speed and direction, moisture content and radiative conditions. Future work could further examine how these effects vary across seasons and under evolving Arctic climate conditions, and could systematically assess how incident flow from different directions interacts with Svalbard's heterogeneous topography and the surrounding ocean and sea ice surfaces."*

- Caption of Figure 3: I would again mention the duration of trajectories (which has a great impact on the data shown) and thus start the caption as "Maximum topography and subsidence for 36-hour backward trajectories...".

- Effectively, not all trajectories go 36 hours backward, because many reach the model domain boundary before. We changed the caption:

- *„The values are averages over trajectories, traced up to 36 hours backward in time, that were initialized at the same time and height level."*

- Caption of Figure 4: here and elsewhere, you mention a Gaussian filter. Can you describe how this was chosen/defined? Additionally, I would specify in the caption that subplots (g,h) relate to ICON_ctrl.

- We added

- *„For visual clarity, a light Gaussian filter ($\sigma$ =0.8) was applied to smooth small-scale features in the contours."*

- The caption of Fig. 4 actually stated that panels (g) and (h) also relate to ICON$_{ctrl}$:

- "Wind speed and direction in $ICON_{ctrl}$ averaged over 14 April 2020 00–12 UTC (a–c, g) and 16 April 2020 00–06 UTC (d–f, h)."

- Lines 172f: I would formulate this a bit more carefully: "The closer match between ICON_ctrl and MOSAiC observations supports the idea that Svalbard's …".

- Changed accordingly.

- Captions of Figs. 7 & 9: I would adjust the last sentence to "Black squares (circles) indicate air parcel positions 6 (12) hours back in time …".

- Changed accordingly.

- Lines 224f: An interesting point. Does Svalbard mostly cause a diversion/delay of the WAI, or also a change in introduced moist static energy into Arctic? Cloud formation / precipitation / diabatic processes around come to mind. Can it be quantified how much the meridional energy flux is reduced by Svalbard's topography? Again, these are just some ideas / general questions I had.

- Thanks for idea! We indeed find that the poleward energy transport is reduced by the topography and added the following text and a supplemental figure:

  - *"Consistent with this, it is found that the topography of Svalbard reduces the poleward energy transport in the wake zone, while slightly increasing it in the flow around Svalbard (Fig. A3a). Averaged over 81–82°N and 0–40° E, the poleward vertically-integrated moist-static energy transport during 14–15 April is up to ∼20 % higher in $ICON_{flat}$ than in $ICON_{ctrl}$ (Fig. A3b–c). The latent heat transport component even increases by up to ∼50% when the topography is removed, but its contribution to the total energy transport is two orders of magnitude smaller than the sensible heat component. Overall, the topography reduces the energy transported into the central Arctic by the WAI."*

- Line 264: It could be argued that putting this section ("3.1.4 Topographic effects on moisture and clouds") before discussing the SEB in 3.1.3, as the difference in cloud structures drives the observed changes. But I think leaving it as-is would also be fine.

- We agree that aspects of this section could be relevant also earlier in the text. However, we want to keep the focus first on the effects on wind and air temperature as the main results.

- Lines 288f: I don't find the abbreviated "25-50%ile", "100%ile" particularly well readable. I would prefer something like "$25^{th}$-$50^{th}$ percentile", "$100^{th}$ percentile".

- Changed accordingly.

- Line 289: Please correct the unit of IVT to "kg m-1 s-1". Please check for correct unit throughout manuscript.

- Changed accordingly.

- Line 311f: You write: "potentially due to the release of more latent heat". Can't you check in ICON whether this is true or not?

- The available output of the ICON climate simulation does not include temperature tendencies, for example, which would be required to assess this robustly. We can argue qualitatively that precipiation, low-level cloud cover and liquid water path on the windward slopes of Svalbard are higher for high vs. low IVT, indicating more condensation and latent heat release:

  - *„This indicates stronger föhn warming in moister poleward advection events, consistent with enhanced condensation and associated latent heat release during windward ascent of the moister air (Elvidge and Renfrew, 2016; Mattingly et al., 2023), as evidenced by higher precipitation, increased low-level cloud cover, and larger cloud liquid water path at high vs. low $IVT_{Sv}$ (Fig. A7d,e,j).“*

- Lines 330ff: You focus on separating cases with high vs. low stability. Are there any other parameters that determine Svalbard's topographic influence?

  - We did indeed address this factor specifically becaus this explained differences between the two situations found for the April 2020 WAI case. However, we discussed several factors throughout the manuscript. We added two sentences at the end of the conclusions to recap this and to suggest aspects that could be addressed in further research:

    - *„This study focused on the April 2020 MOSAiC WAI case and thus on spring conditions with southerly advection, identifying several factors that modulate Svalbard's topographic effects, including static stability, synoptic-scale circulation patterns, wind speed and direction, moisture content and radiative conditions. Future work could further examine how these effects vary across seasons and under evolving Arctic climate conditions, and could systematically assess how incident flow from different directions interacts with Svalbard's heterogeneous topography and the surrounding ocean and sea ice surfaces.“*

- Line 367: You could specify: "Within this wake, Svalbard's topographic effects in the control vs. flat experiment …".

  - Changed accordingly.

- Line 378: You only used the abbreviation "MAM" once. I would leave out defining it, and then here write it out: "March-April-May".

  - Changed accordingly.

- Lines 390ff: I agree that these could be very interesting follow-up studies. Using a coupled model for those could be of great benefit.

  - We added:
    - *„Future studies employing coupled atmosphere-ocean-sea-ice models could help to further quantify these effects.“*